# A Lipidomics Study Reveals Lipid Signatures Associated with Early Allograft Dysfunction in Living Donor Liver Transplantation

**DOI:** 10.3390/jcm8010030

**Published:** 2018-12-29

**Authors:** Hsin-I Tsai, Chi-Jen Lo, Chih-Wen Zheng, Chao-Wei Lee, Wei-Chen Lee, Jr-Rung Lin, Ming-Shi Shiao, Mei-Ling Cheng, Huang-Ping Yu

**Affiliations:** 1Department of Anesthesiology, Chang Gung Memorial Hospital, Taoyuan 333, Taiwan; tsaic79@cgmh.org.tw (H.-I.T.); 8902084@cgmh.org.tw (C.-W.Z.); 2Graduate Institute of Clinical Medical Sciences, Chang Gung University, Taoyuan 333, Taiwan; alanlee@cgmh.org.tw; 3College of Medicine, Chang Gung University, Taoyuan 333, Taiwan; weichen@cgmh.org.tw; 4Metabolomics Core Laboratory, Healthy Aging Research Center, Chang Gung University, Taoyuan 333, Taiwan; chijenlo@mail.cgu.edu.tw; 5Department of General Surgery, Chang Gung Memorial Hospital, Taoyuan 333, Taiwan; 6Department of Liver and Transplantation Surgery, Chang-Gung Memorial Hospital, Chang-Gung University College of Medicine, Taoyuan 333, Taiwan; 7Clinical Informatics and Medical Statistics Research Center and Graduate Institute of Clinical Medicine, Chang Gung University, Taoyuan 333, Taiwan; jr@mail.cgu.edu.tw; 8Department of Biomedical Sciences, Chang Gung University, Taoyuan 333, Taiwan; msshiao@mail.cgu.edu.tw; 9Clinical Metabolomics Core Laboratory, Chang Gung Memorial Hospital at Linkou, Taoyuan 333, Taiwan

**Keywords:** living donor liver transplantation, early allograft dysfunction, lipidomic

## Abstract

Liver transplantation has become the ultimate treatment for patients with end stage liver disease. However, early allograft dysfunction (EAD) has been associated with allograft loss or mortality after transplantation. We aim to utilize a metabolomic platform to identify novel biomarkers for more accurate correlation with EAD using blood samples collected from 51 recipients undergoing living donor liver transplantation (LDLT) by 1H-nuclear magnetic resonance spectroscopy (NMR) and liquid chromatography coupled with mass spectrometry (LC-MS). Principal component analysis (PCA) and orthogonal projection to latent structures-discriminant analysis (OPLS-DA) were used to search for a relationship between the metabolomic profiles and the presence of EAD.Cholesteryl esters (CEs), triacylglycerols (TGs), phosphatidylcholines (PCs) and lysophosphatidylcholine (lysoPC) were identified in association with EAD and a combination of cholesterol oleate, PC (16:0/16:0), and lysoPC (16:0) gave an optimal area under the curve (AUC) of 0.9487 and 0.7884 in the prediction of EAD and in-hospital mortality, respectively after LDLT. Such biomarkers may add as a potential clinical panel for the prediction of graft function and mortality after LDLT.

## 1. Introduction

Over the past few decades, liver transplantation has been a lifesaving treatment for individuals with end stage liver disease and acute liver failure [1]. Because of the scarcity of available organs, extended criteria donors (ECD) are now routinely used around the world for recipients who urgently require liver transplantation. In Asia, where cadaveric donation has been uncommon, living donor liver transplantation (LDLT) has become widely accepted, achieving an impressive 1-year graft and recipient survival rate of nearly 90%, similar to that of deceased donor liver transplantation [2,3]. However, initial poor function of a liver allograft after liver transplantation, termed early allograft dysfunction (EAD), has been associated with allograft loss or mortality after transplantation [4]. A recent cohort study has reported an incidence of EAD of approximately 25% after LDLT, and the risk of graft failure at 90 days was 5.2 times higher in recipients with EAD than for those without. Graft quality, preoperative bilirubin, portal reperfusion pressure, donor age and donor body mass index have been identified as risk factors associated with EAD [5,6]. Although no unanimous definitions have been used to characterize EAD, for this study we adopted the most commonly used and recently validated version [7,8], in which the diagnosis of EAD was clinically based on the levels of serum transaminases, total bilirubin and the international normalized ratio within the first postoperative week.

As EAD may have a negative impact on recipient and graft survival, researchers have attempted to identify new biomarkers, hoping to accurately assess graft function after liver transplantation and initiate timely therapeutic intervention when necessary. Since the liver is a multi-functional organ responsible for the synthesis and catabolism of small and complex molecules, metabolomics has given scientists an opportunity to closely examine metabolites that may help to predict graft viability and function and even patient survival after liver transplantation. Earlier case studies utilizing ^1^H-nuclear magnetic resonance (NMR) spectroscopy have found that elevated levels of circulating uric acid and amino acids, such as glutamine and methionine, in addition to low concentrations of fatty acids as a result of ischemia-reperfusion conditions and a disturbance in the metabolism of fatty acids and amino acids, respectively [9,10] are associated with graft dysfunction. More recent studies using mass spectrometry have also proposed potential biomarkers that may predict poor liver function. Researchers have found that disturbed levels of bile acids, lysophospholipids, phospholipids, sphingomyelins and histidine metabolism products were associated with EAD in donors after brain death (DBD) and in patients with decompensated cirrhosis with increased mortality and severity of the disease [11,12]. In fact, a distinctive metabolic profile was demonstrated in grafts from donation after circulatory death (DCD) and DBD. In connection with early allograft dysfunction and primary non-function, DCD grafts appeared to show higher concentrations of lysophosphatidylcholines (lysoPCs) and increased levels of circulatory tryptophan and kynurenine at the pre-transplant stage but lower concentrations of lysoPCs at the post-transplant stage [13,14]. Ceglarek et al. [15] have also demonstrated that low esterification of plant sterols in recipients after deceased donor liver transplantation was associated with a high risk of EAD. We therefore hypothesized that the recovery of liver function after liver transplantation may be linked to a disturbance in the lipid homeostasis, and in this study, we examine the lipidomic profiles in association with the development of EAD in recipients of LDLT.

## 2. Materials and Methods

### 2.1. Patients

The study received prior approval from the Institutional Review Board of Chang Gung Memorial Hospital (IRB 103-5859A3) and registered under The Australian New Zealand Clinical Trials Registry, ID number: ACTRN12615000446561. Informed consent was obtained from all subjects. A total of 53 recipients undergoing LDLT between May 2015 and October 2016 at Chang Gung Memorial Hospital (Taoyuan, Taiwan) were consecutively recruited to the study. Exclusion criteria included a concurrent septic or shocked status, an anticipated pulmonary hypertension with a preoperative pulmonary wedge pressure greater than 35 mmHg or refusal to provide informed consent. Figure 1 shows a flowchart of the patient selection, allocation and analysis. Of the 53 recipients, 19 had hepatocellular carcinoma (HCC) as the main indication for LDLT, while the other 34 had indications other than HCC, such as alcoholism, hepatitis B or C virus-related cirrhosis, among which two patients were excluded from the metabolomic analysis due to mortality within the first week postoperatively, leaving 51 recipients for metabolomic analysis. From December 2016 to January 2018, a further validation cohort of 21 patients hospitalized for LDLT was recruited as shown in Figure 1.

On the day of transplantation, general anesthesia was performed perioperatively, and allograft implantation started with an anastomosis of the hepatic vein of the graft to the inferior vena cava of the recipient, followed by the reconstruction of the portal vein between the portal vein of the graft and the portal trunk of the recipient. After graft reperfusion, the hepatic arteries of the donor and recipient were reconstructed. Finally, duct-to-duct reconstruction between the hepatic duct of the graft and the common bile duct of the recipient was performed [16]. Cold ischemia time (CIT) was defined as the time from infusion of the cold preservation solution until implantation of the liver allograft in the recipient. Warm ischemia time (WIT) was defined as the time from the start of the hepatic vein reconstruction to the portal reperfusion. The liver transplant clinical outcomes were classified as EAD for grafts meeting one or more of the following criteria [7]: bilirubin ≥10 mg/dL on postoperative day 7, international normalized ratio (INR) ≥1.6 on postoperative day 7, or alanine (ALT) or aspartate (AST) aminotransferase >2000 IU/mL within the first 7 postoperative days. The length of hospital stay, one month mortality and in-hospital mortality were recorded for all recipients.

### 2.2. Blood Samples

Blood samples were obtained after collecting arterial blood from a peripherally indwelling arterial catheter in EDTA (ethylenediaminetertraacetic acid) tubes (BD Vacutainer, Franklin Lakes, NJ, USA) at two time points: T1—before the induction of general anesthesia and T6—on postoperative day 7. The blood was centrifuged immediately at 1000 *g*, 4 °C, for 10 min to obtain plasma. Samples were stored at −80 °C until analysis. The routine biochemical data were measured by the clinical laboratory within the hospital.

### 2.3. NMR Analysis of The Plasma

Before the NMR analysis, the frozen plasma samples were thawed on ice. Next, 350 μL of the plasma sample was mixed with 350 μL of a plasma buffer solution (75 mM Na_2_HPO_4_, 0.08% TSP, 2 mM NaN_3_, 20% D_2_O), and the mixed sample was centrifuged at 12,000 *g* at 277 K for 5 min. Finally, 600 μL of the supernatant was transferred to 5 mm NMR tubes for analysis [17] (Appendix A).

### 2.4. Liquid Chromatography Coupled with Mass Spectrometry (LC-MS)-Based Lipidomic

To 10-μL plasma, 490 μL isopropanol (IPA) (precooling at −20 °C) was added. The mixture was vortexed for 60 s, let stand on ice 30 min, and centrifuged at 12,000 rpm at 4 °C for 30 min. The supernatant was collected in a separate glass tube, diluted with IPA/acetonitrile (ACN)/H_2_O (2/1/1) and centrifuged at 12,000 rpm for 30 min. The clear supernatant was collected for liquid chromatography coupled with mass spectrometry (LC-MS) analysis [18] (Appendix A).

### 2.5. Ultra-Performance Liquid Chromatography (UPLC)-Based Amino Acid Measurement

The plasma samples were collected and stored at −80 °C until assayed. The plasma samples (100 µL) were precipitated by adding an equal volume (100 µL) of 10% sulfosalicylic acid containing an internal standard (norvaline 200 µM) [19]. After protein precipitation, the samples were vortexed and centrifuged at 12,000 g for 10 min at room temp. After the samples were centrifuged, 20 µL of the supernatant was mixed with 60 µL working buffer (borate buffer, pH 8.8). The derivatization was initiated by adding 20 µL of 10 mM AQC (6-aminoquinoly-*N*-hydroxysuccinimidyl carbamate) in acetonitrile. After 10 min incubation, the reactant was mixed with an equal volume of Eluent A (20 mM ammonium formate/0.6% Formic acid/1% acetonitrile) and analyzed using the ACQUITY UPLC System. The AQC derivatization reagent was obtained from the Waters Corporation (Milford, MA, USA) [20] (Appendix A).

### 2.6. Statistical Analysis

The continuous variables data were presented as the mean ± standard deviation (SD), and the independent sample *t*-test and the Mann-Whitney U test were used for the comparison of the EAD and non-EAD groups. The categorical data were presented as frequencies and compared using the chi-square test or the Fisher’s exact test. To identify an independent predictor of postoperative early allograft dysfunction, linear logistic regression analysis was performed. A receiver operator characteristic curve (ROC) was constructed, and the area under the ROC was used to measure the predictive accuracy and to compare between both groups. A *p*-value < 0.001 was considered to be statistically significant. All analyses were performed in R 3.3.2 [21] and SAS 9.4 (SAS Institute, Cary, NC, USA). The metabolomics analysis was performed using several software programs. To maximize the identification of differences in the metabolic profiles between the groups, the orthogonal projection to latent structures-discriminant analysis (OPLS-DA) model was applied and performed using SIMCA-P software (version 8.0, Umetrics, Sweden). The variable importance in the projection (VIP) value of each variable in the model was calculated to indicate its contribution to the classification. The VIP values of those variables greater than 1.0 are considered to be significantly different.

## 3. Results

### 3.1. Patient Characteristics

Table 1 summarized the demographics of the 51 recipients. The 51 recipients had a mean age of 56.10 years (56.10 ± 8.06), with 30 males and 21 females undergoing LDLT. Of the 51 recipients, 12 developed EAD, and 39 had uneventful recoveries during the first week postoperatively. The Biological Model for End-Stage Liver Disease (MELD) score was calculated on the day of transplantation in all cases. No statistical differences in the preoperative MELD score was observed between the EAD and the non-EAD recipients. Although no statistical difference was observed in the time for major blood vessel anastomosis between the two groups, EAD group appeared to have greater blood loss intraoperatively, thus requiring more red blood cell transfusion. Of the 51 recipients, 6 in-hospital mortalities were observed, in which 4 fell in the EAD group while the other 2 in the non-EAD group. Twelve recipients had graft recipient weight ratio (GRWR) <0.8 [22,23], satisfying the definition for small for graft size (SFGS); however, none of them developed EAD within the first week postoperatively or had in-hospital mortality.

The recipients’ clinical performance preoperatively and day 7 postoperatively was illustrated in Table 2. The mean concentrations of routine liver function tests, renal function tests, complete blood cell counts and coagulation profiles, with respect to the two time points (T1—preoperative and T6—on postoperative day 7), were shown in Table 2 and Table 3, including four different *p*-values. The *p*-value denoted as EAD was used to compare EAD patients between T1 and T6, whereas the *p*-value denoted as denoted as non-EAD was used to compare non-EAD patients between T1 and T6. The *p*-value denoted as T1 was used to compare the EAD and non-EAD groups at T1 while the *p*-value denoted as T6 was used to compare the EAD and non-EAD groups at T6. No statistical difference in the preoperative bilirubin, INR and aminotransferases was observed between the EAD and non-EAD groups; however, on postoperative day 7 (T6), the EAD group showed a statistically significant deterioration of liver function, renal function and INR. Not surprisingly, non-EAD patients showed a noticeable improvement in INR and bilirubin.

### 3.2. Change in Circulatory Amino Acid Profiles in Recipients with EAD

In Table 3, amino acids appear to remain relatively stable in concentrations regardless the EAD or non-EAD status postoperatively. We observed that although the concentrations of branched chain amino acids (BCAA) and aromatic amino acids (AAA) showed no significant difference between the EAD and the non-EAD groups, the Fischer ratio (BCAA/AAA) at T6, and of particular interest, showed a significant increase in the non-EAD group.

### 3.3. Changes in NMR Plasma Profiles in Recipients with EAD

Figure 2 demonstrates the ^1^H NMR plasma profiles of the EAD and non-EAD patients on postoperative day 7 (T6). The score plot is displayed in Figure 2A (*R*^2^X = 0.691, *R*^2^Y= 0.521, *Q*^2^ = 0.219) while OPLS-DA loading coefficient plot showing that the difference between the EAD and non-EAD groups is shown in Figure 2B. The positive signals (low density lipoprotein (LDL)) and lipid relative signals) correspond to the increased metabolites in the plasma from the EAD group.

### 3.4. Changes in Circulatory Lipid Profiles in Recipients with EAD

To discover relevant lipid features of the recipients, blood samples obtained at T6 were analyzed by ultra performance liquid chromatography - time of flight mass spectrometry (UPLC-TOFMS) in electrospray positive ion mode. Three replicates were performed, and an OPLS-DA model was built. As shown in Figure 3A, the model revealed lipid differences, with a clear separation of metabolites between the EAD and non-EAD groups at T6 (*R*^2^X = 0.895, *R*^2^Y = 0.965, *Q*^2^ = 0.893).

From this model, 22 features were selected that differentiated between the two groups with variables importance for the projection or VIP >1.0 and *p*< 0.001. These 22 features were represented according to lipid categories included 6 cholesterols, 1 triacylglycerols (TGs), 8 phosphatidylcholines (PCs), 4 lysophosphatidylcholines (lysoPC), 1 L-acetylcarnitine, and 2 unknown metabolites as shown in Table 4.

### 3.5. Discriminative Ability of Potential Biomarkers for EAD and In-Hospital Mortality

The distribution of CE, PC, and lysoPC in EAD and non-EAD were evaluated. The Mann-Whitney test revealed that the amount of cholesterol linoleate, cholesterol oleate and lysophosphatidylcholines were significantly lower in the EAD group while phosphatidylcholines were higher in the EAD group than in the non-EAD group (Table 4). As 22 metabolites were identified with the ability to discriminate patients with EAD from those without EAD, we evaluated their ability to serve as potential metabolic predictors for EAD. The areas under the curve (AUC) of the receiver-operating characteristics (ROC) curve of 3 selected metabolites and 4 clinical parameters (total bilirubin, AST, ALT and INR) are summarized in Table 5.

The predictive ability of metabolite cholesterol oleate, lysoPC (16:0), PC (16:0–16:0) evaluated by ROC was 0.9338, 0.8921 and 0.8376, respectively, all of which higher than that of INR, AST or ALT alone. A combination of cholesterol oleate, PC (16:0/16:0), and lysoPC (16:0) enhanced the predictive value (Figure 4A) to an optimal AUC of 0.9487 in distinguishing non-EAD from EAD. These findings suggested that these lipidomics-derived biomarkers may have the potential to identify patients at high risk for EAD. Even though cholesterol oleate, lysoPC (16:0), PC (16:0–16:0) showed an AUC of 0.6316, 0.8125 and 0.6530 comparable to that of individual clinical markers in predicting a long hospital stay of more than 45 days, the combination gave an AUC of 0.8207 as shown in Figure 4B. Similar ROCs were constructed and the metabolite combination gave an AUC of 0.7884 in the prediction of in-hospital mortality as depicted in Figure 4C.

### 3.6. External Validation of Lipidomic Profiling as Prediction of EAD, Long Hospital Stay and In-Hospital Mortality

In a separate cohort of 21 patients undergoing LDLT, the three metabolites were used to validate the outcomes in terms of the occurrence of EAD, hospital stay and mortality (See Appendix A). The demographic and biochemical data were shown in Appendix A and S2, respectively. The ROC curve analysis for individual metabolite was shown in Table 5 and Figure 5A. The combination cholesterol oleate, lysoPC (16:0) and PC (16:0–16:0) gave an excellent prediction of EAD with an AUC of 0.9722, a value much higher than total bilirubin, AST, ALT or INR alone. The combination also gave an AUC of 0.8454 and 1.000 in the prediction of long hospital stay and in-hospital mortality as shown in Figure 5B,C.

### 3.7. Schematic Illustration of Metabolite Alternation in EAD after LDLT

A schematic illustration of the lipid profile alteration was proposed in Figure 6. A decreased level of CE and lysoPC and an increased level of phospholipids were seen in EAD and demonstrated in the figure.

## 4. Discussion

In the present study, we have examined the metabolomic differences between EAD and non-EAD patients undergoing LDLT despite their comparable preoperative and perioperative parameters. On postoperative day 7, recipients developing EAD demonstrated a statistically significant deterioration in their coagulation profiles and renal and liver functions compared to non-EAD recipients. Further in-depth metabolomic studies revealed disturbances in the distribution of amino acid and lipids, providing potential biomarkers for the early detection of EAD.

Specific alterations in the amino acid profiles has been described in patients with liver cirrhosis, including a decrease in branched-chain amino acids (BCAA) and an increase in aromatic amino acids (AAA). Recent literature has demonstrated that a decreased BCAA/AAA ratio or “Fischer ratio” may act as a surrogate assessment of liver function in acute and chronic liver failure [24]. The present study has revealed that on postoperative day 7 (T6), the majority of amino acids showed no significant differences between the EAD and non-EAD groups except for an increase in the Fischer’s ratio in the non-EAD group. Consistent with earlier literature, we have demonstrated amino acid alterations during liver failure and improvements after liver transplantation [25,26]. Amino acids alone could not be used to discriminate between the EAD and non-EAD groups on postoperative day 7 and therefore are poor biomarkers of the outcome in liver transplantation. However, the Fischer ratio may be a potential indicator of liver function after liver transplantation.

A few specific lipids, such as cholesteryl esters, TGs, and phospholipids, could effectively discriminate EAD from non-EAD on postoperative day 7 (T6). In our study, the EAD group had a significantly lower level of long chain lysoPC (16:0) than the non-EAD group on postoperative day 7, demonstrating that a reduction in lysoPC correlates with the severity of liver disease. Cano et al. [27] have recently demonstrated that increased levels of sphingomyelins and PC 16:0/16:0 and 16:0/18:0 may be linked to the severity of liver fibrosis in patients with hepatitis C virus (HCV) one year after transplantation. Similarly, we have demonstrated that specific metabolites, namely, PC 16:0/16:0, 16:0/18:1, 16:1/16:0, increased in the plasma levels in patients within 1 week after LDLT. Decreased biosynthesis of TGs and cholesteryl esters (CEs) was also observed in the EAD patients. Even though the precise mechanisms of each lipid species and their impact on hepatocytes remain to be elucidated, we have shown that a combination of cholesterol oleate, PC (16:0/16:0) and lysoPC (16:0) may be associated with the development of EAD. Clinically, we have also demonstrated that the metabolite combination, with an AUC of 0.7884 and 1.000 in the study and validation group, respectively, may better predict in-hospital mortality than currently available clinical markers, AST, ALT, INR or total bilirubin after LDLT.

The serum bilirubin level may not appear to be a sensitive indicator of hepatic dysfunction as the elevation of serum bilirubin may be multifactorial. Although it has been used as one of the criteria to distinguish EAD from non-EAD patients postoperatively, bilirubin alone does not serve as a good biomarker for outcomes in liver transplantation. A combination of amino acids, bilirubin, and lipids appears to give better prognostic accuracy.

The strength of our study is the prospective presentation of graft function and short-term mortality in patients undergoing LDLT. We have minimized the potential selection bias by excluding patients with known poor prognostic factors in an attempt to keep the recipient cohort as homogenous as possible. However, the relatively small number of patients is one limitation. Although lipidomics appears to be a promising tool in the discrimination of patients with EAD and without EAD, validation in a larger data set is warranted.

## 5. Conclusions

A significant increase in the Fischer ratio is observed in the non-EAD group. Changes in cholesteryl esters, lysophospholipids and phosphatidylcholines may be associated with the development of EAD and serve as prognostic factors for postoperative graft function and patient mortality.

## Figures and Tables

**Figure 1 jcm-08-00030-f001:**
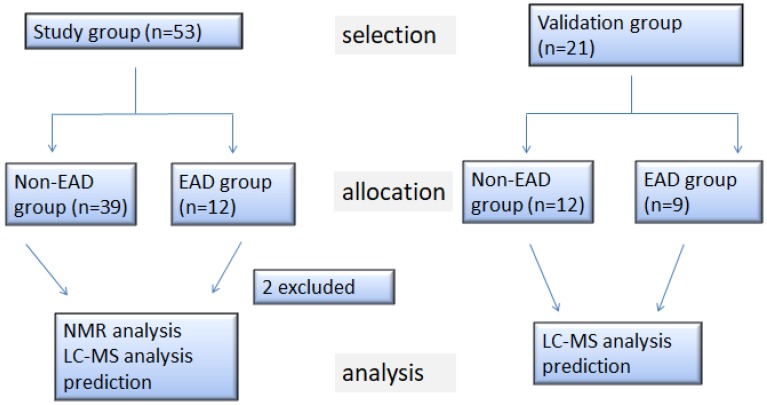
Flow diagram of the patient selection, allocation and analysis. HCC, hepatocellular carcinoma; non-HCC, non-hepatocellular carcinoma; EAD, early allograft dysfunction; NMR, nuclear ^1^H-nuclear magnetic resonance spectroscopy; LC-MS, liquid chromatography coupled with mass spectrometry.

**Figure 2 jcm-08-00030-f002:**
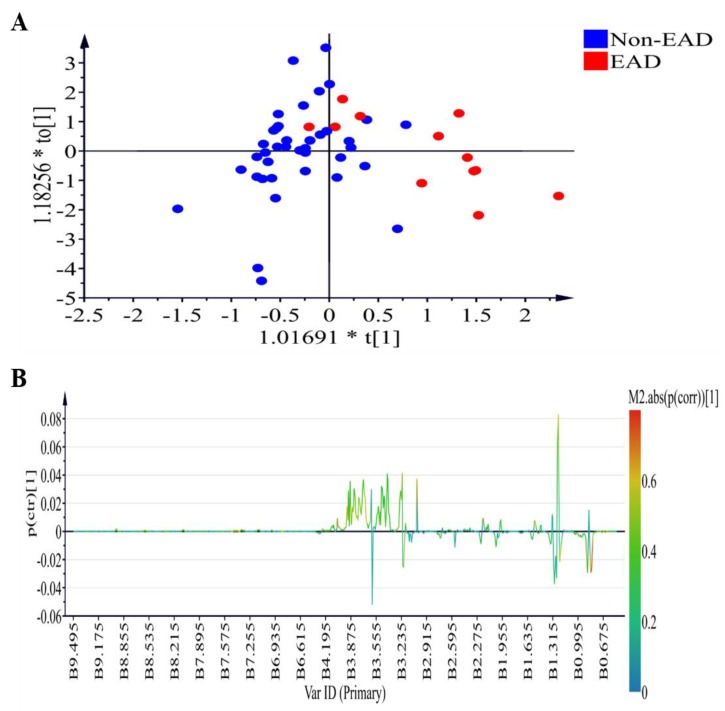
Plasma samples analyzed by ^1^H-NMR in the comparison of the EAD and non-EAD patients on postoperative day 7 (T6). (**A**) OPLS-DA scores plot presenting T6 EAD (red) and T6 non-EAD (blue) with fitness and predictive power *R*^2^X = 0.691, *R*^2^Y = 0.521, *Q*^2^ = 0.219. (**B**) OPLS-DA loading coefficient plot showing that the difference between the EAD and non-EAD groups. ^1^H NMR, ^1^H-nuclear magnetic resonance spectroscopy; OPLS-DA, orthogonal projection to latent structures-discriminant analysis; EAD, early allograft dysfunction.

**Figure 3 jcm-08-00030-f003:**
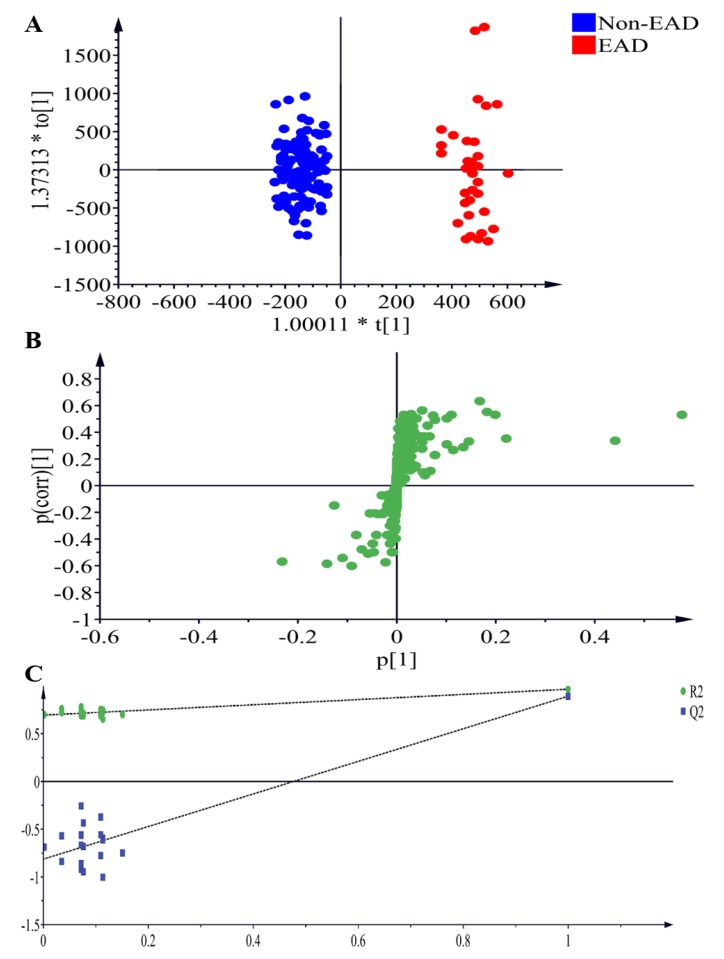
Plasma samples analyzed by LC-MS in electrospray positive ion mode, comparing EAD and nonEAD recipients. (**A**) OPLS-DA scores plot presenting T6 EAD (red) and T6 Non-EAD (blue) with fitness and predictive power *R*^2^X = 0.895, *R*^2^Y = 0.965, *Q*^2^ = 0.893. (**B**) Metabolites with significant differences in electrosrapy positive ion modes between normal EAD and Non-EAD are presented in S-plots. (**C**) Validation of the OPLS-DA model by a class permutation analysis for (**A**). LC-MS, liquid chromatography coupled with mass spectrometry; OPLS-DA, orthogonal projection to latent structures-discriminant analysis; EAD, early allograft dysfunction.

**Figure 4 jcm-08-00030-f004:**
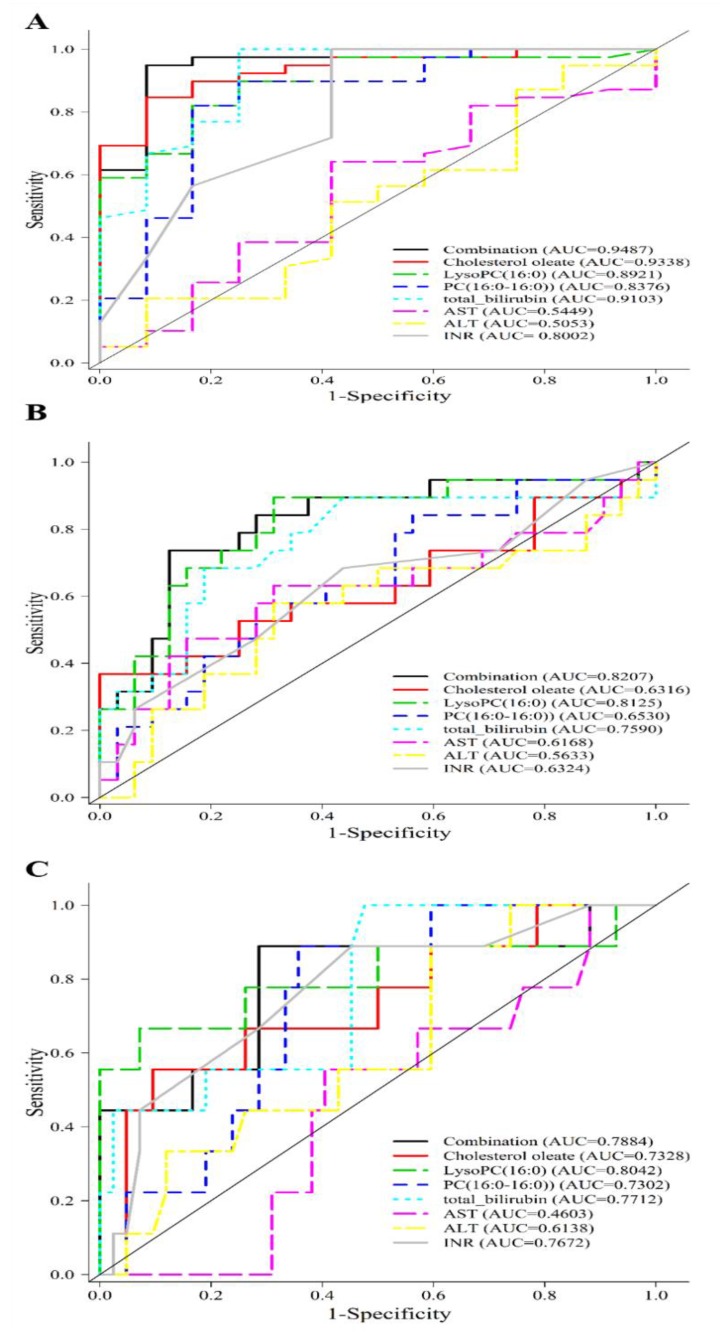
Prediction of early allograft dysfunction, all-cause in-hospital mortality, long hospital stay in study cohort. (**A**) A combination of cholesterol oleate, PC (16:0/16:0), and lysoPC (16:0) metabolites gives an AUC of 0.9487 in the prediction of EAD in the study group. (**B**) A combination of c cholesterol oleate, PC (16:0/16:0), and lysoPC (16:0) metabolites gives an AUC of 0.8207 in the prediction of hospital stay more than 45 days in the study group. (**C**) A combination of cholesterol oleate, PC (16:0/16:0), and lysoPC (16:0) metabolites gives an AUC of 0.7884 in the prediction of in-hospital mortality in the study group.

**Figure 5 jcm-08-00030-f005:**
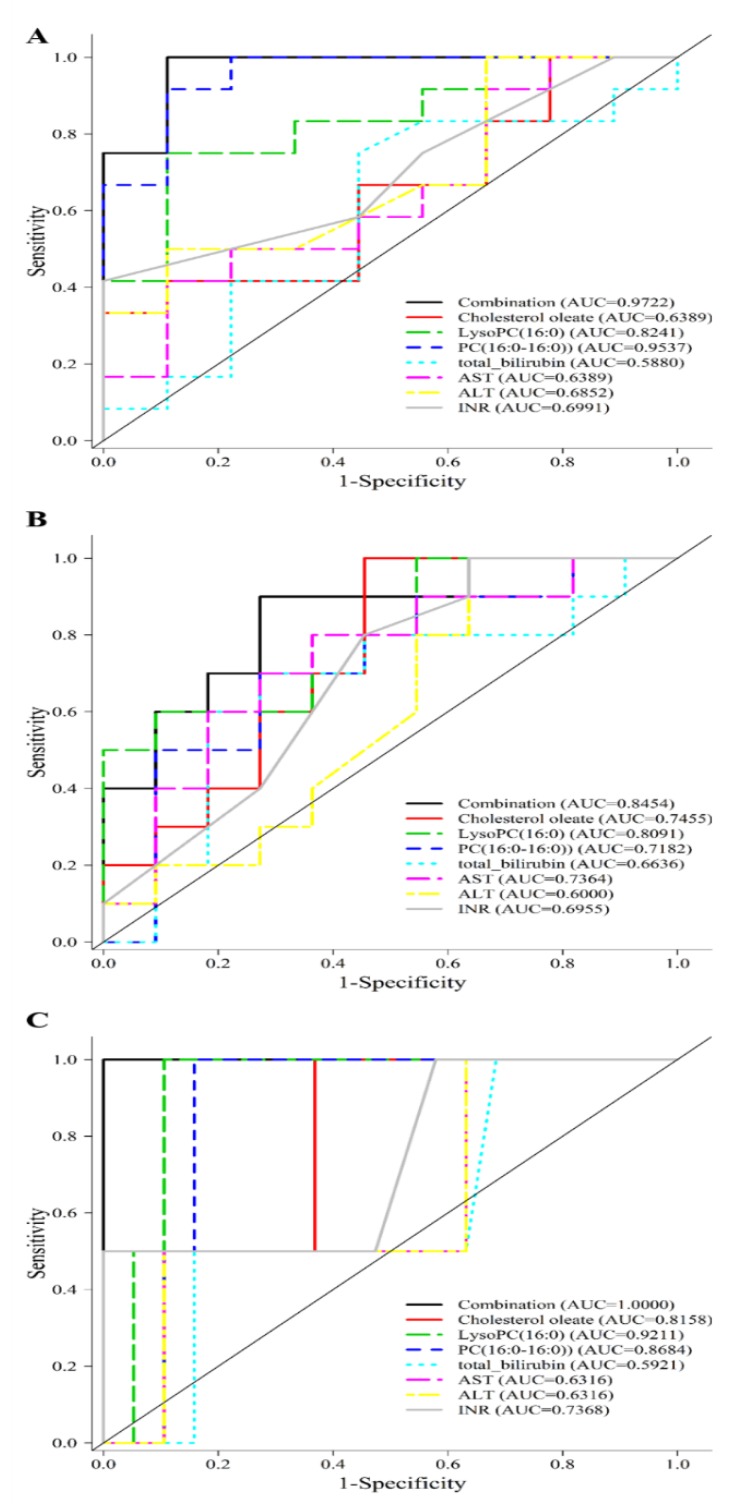
Prediction of early allograft dysfunction, all-cause in-hospital mortality, long hospital stay in validation cohort. (**A**) A combination of cholesterol oleate, PC (16:0/16:0), and lysoPC (16:0) metabolites gives an AUC of 0.7884 in the prediction of in-hospital mortality in the study group. (**B**) A combination of cholesterol oleate, PC (16:0/16:0), and lysoPC (16:0) metabolites gives an AUC of 0.9722 in the prediction of EAD in the validation group. (**C**) A combination of cholesterol oleate, PC (16:0/16:0), and lysoPC (16:0) metabolites gives an AUC of 0.8454 in the prediction of hospital stay more than 45 days in the validation group.

**Figure 6 jcm-08-00030-f006:**
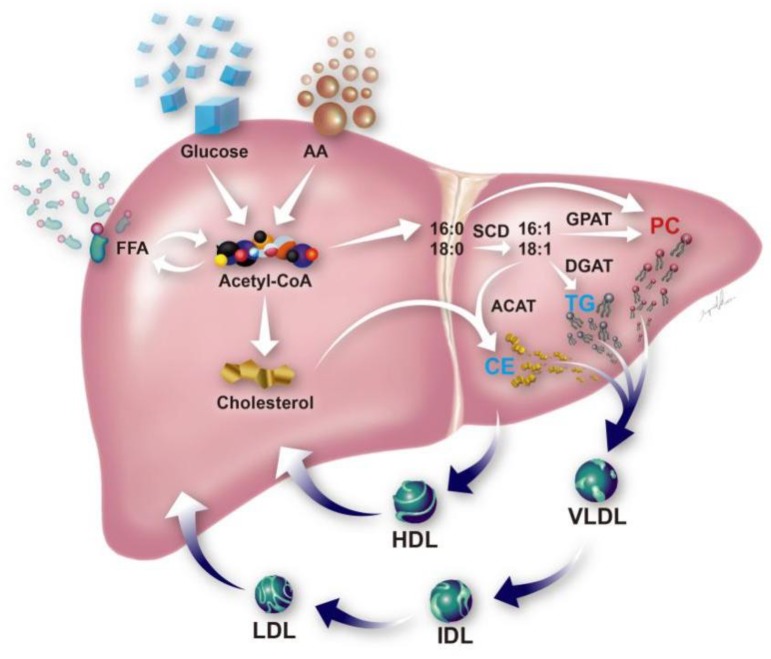
Schematic illustration of metabolic disturbances associated with poor outcomes of liver transplants.Metabolites that increased in the plasma level in early allograft dysfunction are labeled in red (i.e., PC) whereas those that decreased in level are labeled in light blue (i.e., TG and CE). The enzymes involved in the de novo synthesis of TG, PC and CE, including stearoyl-CoA desaturase (SCD), acyl-CoA:cholesterol acyltransferase (ACAT), diacylglycerol acyltransferase (DGAT), and microsomal glycerol phosphate acyltransferase (GPAT), are located in the endoplasmic reticulum membrane. PC, phosphatidylcholine; CE, cholesteryl ester; TG, triacylglycerol. HDL, high density lipoprotein; LDL, low density lipoprotein; VLDL, very low density lipoprotein; IDL, intermediate density lipoprotein; AA, amino acid; FFA, free fatty acid.

**Table 1 jcm-08-00030-t001:** Summary of clinical data for living donor liver transplantation recipients.

	EAD ^11^ (*n* = 12)	Non-EAD ^11^ (*n* = 39)	*p*-Value
Height (m)	1.63 ± 0.09	1.63 ± 0.08	0.99
Weight (kg)	66.43 ± 13.47	67.63 ± 11.66	0.78
BMI ^1^	24.92 ± 13.47	25.31 ± 3.44	0.74
Age (years)	56.42 ± 10.82	56.00 ± 7.30	0.88
Gender (M/F)			0.74
M	8	22	
F	4	17	
Blood type			
A	5	15	1.00
B	2	7	
O	5	15	
AB	0	2	
MELD ^2^	19.92 ± 11.89	16.69 ± 0.09	0.29
HBV ^3^ (Yes/No)	6/6	18/21	0.82
HCV ^4^ (Yes/No)	3/9	12/27	1.00
Alcoholism (Yes/No)	4/8	12/2	1.00
ABO incompatibility (Yes/No)	2/10	6/33	1.00
CKD ^5^ (Yes/No)	5/7	5/34	0.04
Graft (g)	615.00 ± 122.21	644.36 ± 146.22	0.53
GRWR ^6^ (%)	0.96 ± 0.13	0.98 ± 0.30	0.82
Blood loss (mL) PRBC ^7^ (U) FFP ^8^ (U) Platelet (U)	2962.50 ± 2111.13	1705.64 ± 166348	0.04
	13.42 ± 8.97	7.90 ± 7.52	0.04
	13.83 ± 11.07	11.69 ± 11.05	0.56
	12.00 ± 8.86	8.62 ± 9.52	0.28
CIT ^9^ (minutes)	41.50 ± 37.48	52.97 ± 34.94	0.33
WIT ^10^ (minutes)	37.08 ± 7.83	42.31 ± 10.94	
In hospital mortality (Yes/No)	4/8	2/37	0.13
Hospital stay ≥45 days (Yes/No)	9/3	12/27	

^1^ body mass index; ^2^ model for end-stage liver disease; ^3^ hepatitis B virus; ^4^ hepatitis C virus; ^5^ chronic kidney disease; ^6^ graft recipient weight ratio; ^7^ packed red blood cell; ^8^ fresh frozen plasma; ^9^ cold ischemia time; ^10^ warm ischemia time; ^11^ early allograft dysfunction.

**Table 2 jcm-08-00030-t002:** Biochemical data for the patients before and after liver transplantation. The *p*-value denoted as EAD was used to compare EAD patients between T1 and T6, whereas the *p*-value denoted as denoted as non-EAD was used to compare non-EAD patients between T1 and T6. The *p*-value denoted as T1 was used to compare the EAD and non-EAD groups at T1 while the *p*-value denoted as T6 was used to compare the EAD and non-EAD groups at T6.

	T1 EAD ^10^ (*n* = 12)	T1 Non-EAD ^10^ (*n* = 39)	T6 EAD ^10^ (*n* = 12)	T6 Non-EAD ^10^ (*n* = 39)	*p*-Value	*p*-Value	*p*-Value	*p*-Value
					(EAD ^10^)	(Non-EAD ^10^)	(T1)	(T6)
Hemoglobin (g/dL)	9.30 ± 2.05	10.76 ± 2.47	11.20 ± 2.77	10.28 ± 1.39	<0.01	0.29	0.07	0.13
Hematocrit (%)	27.88 ± 6.15	50.38 ± 117.49	31.78 ± 6.97	30.08 ± 4.03	0.02	0.29	0.51	0.29
Platelet (1000/dL)	62.42 ± 42.55	87.94 ± 45.37	40.58 ± 16.43	71.41 ± 37.86	0.08	<0.01	0.09	<0.01
PT ^1^ (sec)	18.59 ± 6.48	19.11 ± 8.52	18.52 ± 4.89	14.05 ± 1.64	0.98	<0.01	0.85	<0.01
INR ^2^	1.60 ± 0.61	1.67 ± 0.74	1.62 ± 0.43	1.23 ± 0.14	0.94	<0.01	0.78	<0.01
APTT ^3^ (sec)	40.92 ± 19.55	36.43 ± 12.16	45.77 ± 19.94	31.50 ± 7.80	0.43	0.03	0.34	<0.01
Total protein (g/dL)	6.18 ± 1.42	6.66 ± 0.90	4.13 ± 0.43	4.33 ± 0.43	<0.01	<0.01	0.17	<0.01
Albumin (g/dL)	3.02 ± 0.73	3.05 ± 0.72	2.53 ± 0.29	2.63 ± 0.31	0.05	<0.01	0.90	0.17
BUN ^4^ (mg/dL)	39.17 ± 40.10	17.27 ± 17.59	53.78 ± 41.17	23.22 ± 12.13	0.07	<0.01	<0.01	0.37
Creatinine (mg/dL)	1.91 ± 2.20	1.08 ± 1.49	1.99 ± 1.32	0.78 ± 0.50	0.87	0.18	0.14	<0.01
GFR ^5^(mL/mL/m/1.73 min/1.73 m^2^)	81.42 ± 60.10	114.00 ± 53.55	65.50 ± 79.17	116.36 ± 47.78	0.27	0.78	0.08	<0.01
Sodium (mEq/L)	140.42 ± 7.17	138.43 ± 3.92	123.98 ± 35.50	136.92 ± 3.03	0.15	0.07	0.22	<0.01
Potassium (mEq/L)	3.61 ± 0.55	4.63 ± 5.51	3.77 ± 0.67	3.71 ± 0.50	0.50	0.31	0.53	0.03
Calcium (mg/dL)	8.39 ± 0.40	8.26 ± 0.57	7.62 ± 0.75	7.43 ± 0.49	0.01	<0.01	0.47	0.74
Sugar (mg/dL)	140.92 ± 45.05	132.08 ± 48.87	219.92 ± 78.30	179.74 ± 55.22	<0.01	<0.01	0.58	0.32
LDH ^6^ (U/L)	299.58 ± 203.00	245.10 ± 79.19	439.67 ± 236.66	328.15 ± 129.62	0.111	<0.01	0.171	0.04
Total bilirubin (mg/dL)	10.44 ± 16.24	4.25 ± 5.85	13.99 ± 8.96	2.64 ± 2.24	0.360	0.071	0.048	<0.01
Direct bilirubin (mg/dL)	6.02 ± 10.14	2.03 ± 2.98	8.14 ± 5.57	1.51 ± 1.43	0.363	0.241	0.032	<0.01
AST ^7^ (U/L)	123.50 ± 189.12	68.90 ± 19.08	109.58 ± 79.96	119.90 ± 84.13	0.811	<0.01	0.112	0.71
ALT ^8^ (U/L)	52.00 ± 37.67	34.87 ± 19.08	191.58 ± 134.26	214.56 ± 181.85	<0.01	<0.01	0.039	0.69
ALP ^9^ (U/L)	95.83 ± 44.70	108.64 ± 53.20	127.00 ± 65.96	88.31 ± 49.30	0.154	0.040	0.454	0.03

^1^ prothrombin time; ^2^ internationalized ratio; ^3^ activated partial thromboplastin time; ^4^ blood urea nitrogen; ^5^ glomerular filtration rate; ^6^ lactate dehydrogenase; ^7^ aspartate aminotransferase; ^8^ alanine aminotransferase; ^9^ alkaline phosphatase; ^10^ early allograft dysfunction.

**Table 3 jcm-08-00030-t003:** Concentrations of amino acid of EAD in comparison to non-EAD recipients at T6 (postoperative day 7).

	T6 EAD ^3^ (*n* = 12)	T6 non-EAD ^3^ (*n* = 39)	*p*-Value
Histidine (μM)	110.6 ± 38.7	78.8 ± 15.4	0.0166
Asparagine (μM)	59.8 ± 17.2	65.3 ± 18.0	0.3511
Taurine (μM)	47.0 ± 28.5	39.7 ± 15.0	0.4101
Serine (μM)	111.0 ± 53.7	116.6 ± 29.7	0.7349
Glutamine (μM)	595.8 ± 250.5	557.4 ± 97.3	0.6135
Arginine (μM)	90.6 ± 48.9	90.7 ± 23.8	0.9935
Glycine (μM)	333.5 ± 157.9	275.9 ± 131.9	0.2129
Citrulline (μM)	41.4 ± 24.8	32.9 ± 8.9	0.2664
Glutamate (μM)	64.6 ± 34.6	71.9 ± 32.7	0.5043
Threonine (μM)	149.8 ± 69.7	172.7 ± 49.7	0.2114
Alanine (μM)	264.5 ± 159.7	246.3 ± 92.6	0.7131
Proline (μM)	237.2 ± 174.0	176.5 ± 61.8	0.2590
Ornithine (μM)	57.9 ± 33.5	58.3 ± 17.5	0.9735
Cystine (μM)	64.2 ± 35.4	51.5 ± 20.5	0.2594
Lysine (μM)	208.2 ± 72.6	246.6 ± 56.6	0.0607
Tyrosine (μM)	82.6 ± 29.9	68.9 ± 24.4	0.1133
Methionine (μM)	41.3 ± 37.6	34.6 ± 16.2	0.5573
Valine (μM)	201.5 ± 52.7	229.2 ± 61.8	0.1670
Isoleucine (μM)	63.4 ± 18.8	72.6 ± 26.7	0.2728
Leucine (μM)	91.4 ± 24.4	110.8 ± 40.6	0.1239
Phenylalanine (μM)	97.2 ± 36.9	73.2 ± 18.5	0.0498
Tryptophan (μM)	51.0 ± 21.4	43.0 ± 13.1	0.2399
Essential AA (μM)	1046.0 ± 295.1	1087.4 ± 234.7	0.6174
BCAA ^1^ (μM)	356.3 ± 90.7	412.6 ± 126.2	0.1587
AAA ^2^ (μM)	230.8 ± 80.3	185.1 ± 47.1	0.0827
Fischer’s ratio	1.7 ± 0.5	2.3 ± 0.5	0.0010

^1^ aromatic amino acids; ^2^ branched chain amino acids; ^3^ early allograft dysfunction.

**Table 4 jcm-08-00030-t004:** A List of metabolites that discriminated the EAD ^1^ from the non-EAD ^1^ groups (VIP > 1.0 & *p* < 0.001).

Metabolites	Adduct	EAD (*n* = 12)	Non-EAD (*n* = 39)	*p*-Value	VIP Value
PC ^2^ (16:0–18:1)	M + H	293,234.70 ± 88,445.42	190,835.04 ± 64,727.83	8.80 × 10^−8^	14.87
Cholesterol linoleate		8247.39 ± 9123.72	23,705.64 ± 10,062.66	1.41 × 10^−13^	5.94
PC (18:1–18:1)	M + H	109,562.18 ± 32,521.47	86,965.79 ± 23,297.41	0.0004	5.71
PC (16:1–16:0)	M + H	19,371.66 ± 14,900.85	7182.02 ± 5130.27	3.08 × 10^−5^	5.13
PC (36:4)	M + H	21,621.80 ± 8897.53	11,880.11 ± 5475.14	2.51 × 10^−7^	4.67
PC (16:0–16:0)	M + H	13,302.07 ± 5827.96	6006.07 ± 3085.78	1.33 × 10^−8^	4.33
PC (16:0–18:2)	M + H	42,095.65 ± 15,600.17	31,631.49 ± 11,841.82	0.0006	3.76
Cholesterol linoleate	M + NH_4_	1768.35 ± 3256.70	7312.47 ± 3399.90	1.36 × 10^−14^	3.62
LysoPC ^3^ (16:0)	M + H	1245.59 ± 1282.79	4852.28 ± 2746.80	1.52 × 10^−19^	2.79
PC (18:1–18:0)	M + H	9301.80 ± 3127.75	5999.75 ± 2319.29	6.12 × 10^−7^	2.57
Cholesterol oleate		1655.40 ± 934.69	3933.22 ± 1373.84	2.79 × 10^−18^	2.38
TG ^4^ (52:3)	M + Na	15,257.06 ± 3186.91	18,383.95 ± 3284.08	1.77 × 10^−6^	2.20
Met0295		2752.32 ± 2798.94	774.76 ± 766.19	0.0002	2.00
Met0592		3071.09 ± 1361.53	1429.76 ± 1076.40	7.83 × 10^−12^	1.87
LysoPC (18:2)	M + H	764.32 ± 840.10	2425.17 ± 1466.14	2.32 × 10^−13^	1.78
L–Acetylcarnitine	M + H	2265.22 ± 2044.40	849.66 ± 824.14	0.0003	1.61
Cholesterol arachidonate		227.70 ± 354.57	1312.62 ± 883.71	7.51 × 10^−20^	1.50
Cholesterol		1745.76 ± 640.41	994.24 ± 411.78	5.96 × 10^−8^	1.32
LysoPC (18:1)	M + H	392.86 ± 576.23	1254.31 ± 834.29	9.18 × 10^−10^	1.23
LysoPC (18:0)	M + H	139.32 ± 229.46	876.64 ± 626.57	1.03 × 10^−19^	1.22
Cholesterol linoleate	M + Na	267.63 ± 856.69	1036.13 ± 854.38	6.75 × 10^−6^	1.06
PC (32:1)	M + Na	559.23 ± 782.39	44.72 ± 119.20	0.0005	1.02

^1^ early allograft dysfunction; ^2^ phosphatidylcholine; ^3^ lysophosphatidylcholines; ^4^ triacylglycerol.

**Table 5 jcm-08-00030-t005:** Receiver operating characteristic (ROC) curve analysis for individual metabolites in study and validation group.

Metabolites	Study Group		Validation Group	
	AUC ^6^	Standard Error	AUC ^6^	Standard Error
Cholesterol oleate	0.9338	0.0350	0.6389	0.1271
LysoPC ^1^ (16:0)	0.8921	0.0491	0.8241	0.0942
PC ^2^ (16:0–16:0)	0.8376	0.0767	0.9537	0.0428
Total bilirubin	0.9103	0.0530	0.5880	0.1365
INR ^3^	0.8002	0.0823	0.6991	0.1149
ALT ^4^	0.5053	0.1011	0.6852	0.1218
AST ^5^	0.5449	0.0988	0.6699	0.1280

^1^ lysophosphatidylcholines; ^2^ phosphatidylcholine; ^3^ internationalized ratio; ^4^ alanine aminotransferase; ^5^ aspartate aminotransferase; ^6^ area under the curve.

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
