# Peer review of "A Lipidomics Study Reveals Lipid Signatures Associated with Early Allograft Dysfunction in Living Donor Liver Transplantation"

_jcm, 2018, doi:10.3390/jcm8010030_

Round 1
Reviewer 1 Report
This is an interesting study examining lipid signatures associated with early allograft dysfunction in LDLT.
You have referenced the key articles that have lead you to perform this study, but your findings that LysoPC (16:0) and Lyso (18:0) were identified in association with EAD needs more discussion. Your study demonstrates that a reduction in LysoPC (16:0) correlates with the severity of liver disease whereas your reference # 14 Xu et al. found the exact opposite. They noted that LysoPC (16:0) and LysoPC(18:0) were more abundant in liver recipients undergoing EAD (p<0.05). Like your study this was an exploratory study using lipidomics to try and find a biomarker of EAD but came up with an opposite result. This does deserve some comment in your discussion, especially as you have referenced their study..
Author Response
Response Letter
Reviewer 1. You have referenced the key articles that have lead you to perform this study, but your findings that LysoPC (16:0) and Lyso (18:0) were identified in association with EAD needs more discussion. Your study demonstrates that a reduction in LysoPC (16:0) correlates with the severity of liver disease whereas your reference # 14 Xu et al. found the exact opposite. They noted that LysoPC (16:0) and LysoPC(18:0) were more abundant in liver recipients undergoing EAD (p<0.05). Like your study this was an exploratory study using lipidomics to try and find a biomarker of EAD but came up with an opposite result. This does deserve some comment in your discussion, especially as you have referenced their study.
Response:
Thank you for your suggestion. In reference #14, authors have collected tissues from DBD (donation after brain death) and DCD (donation after circulatory death) donors. There was no obvious difference in the DBD group from pre- to post-transplant; all lipids were more abundant in the DCD at pre-transplant stage compared with DBD. In the DCD group, lysoPC 16:0) and lysoPC (18:0) showed lower concentration at post-transplant stage. Also, they have found the two lysoPCs at pre-transplant were higher in the EAD group, suggesting that lysoPC (16:0) and lysoPC (18:0) have a role in signalling liver tissue damage before transplantation. In our study, decreased levels of lysoPC (16:0) and lysoPC (18:0) were found 7 days after transplantation, in association of the development of EAD. I apologize that I should have explained more clearly in the manuscript. I have revised the paragraph as highlighted in the manuscript that read “DCD grafts appeared to show higher concentrations of lysophosphatidylcholines (lysoPCs) and increased levels of circulatory tryptophan and kynurenine at pre-transplant stage but lower concentrations of lysoPCs at post-transplant stage.” (Please see page 2).

Reviewer 2 Report
The author try to correlate some lipids from blood samples to early allograft dysfunction in patient that undergone liver transplant. The main finding appears to be a significant decrease in Fisher's ratio found as early as 7 days after transplantation in early allograft dysfunction patients. The finding of an early predictor for graft dysfunction is of great importance in this field and these findings might have some relevance if confirmed in future larger studies.
In fact, the main concern is about the number of patients involved that is too little.
There are minor revisions too:
In Table 2A four p-values are shown and their meaning are explained in the text; however, the explanation should be added also in the figure caption. In the results section and possibly in figure caption, the results referred to table 2A, especially the significant ones, should be described in more detail.
Generally speaking, figure captions are poorly written. A reader should be able to understand perfectly a table or graph only reading the figure caption.
The discussion should be more focused on the topic. I suggest to reduce the parts that are off topic and reinforce those that regard the main topic of the paper.
Author Response
Response Letter
Reviewer 2. In Table 2A four p-values are shown and their meaning are explained in the text; however, the explanation should be added also in the figure caption. In the results section and possibly in figure caption, the results referred to table 2A, especially the significant ones, should be described in more detail.
Generally speaking, figure captions are poorly written. A reader should be able to understand perfectly a table or graph only reading the figure caption.
The discussion should be more focused on the topic. I suggest to reduce the parts that are off topic and reinforce those that regard the main topic of the paper.
Response:
Thank you for your kind comments. The figure captions have been revised as suggested.
Table 1. Summary of clinical data for living donor liver transplantation recipients
Table 2A. Biochemical data for the patients before and after liver transplantation. The p-value denoted as EAD was used to compare EAD patients between T1 and T6, whereas the p-value denoted as denoted as non-EAD was used to compare non-EAD patients between T1 and T6. The p-value denoted as T1 was used to compare the EAD and non-EAD groups at T1 while the p-value denoted as T6 was used to compare the EAD and non-EAD groups at T6
Table 2B. Concentrations of amino acid of EAD in comparison to non-EAD recipients at T6 (postoperative day 7).
Figure 2. Plasma samples analyzed by 1H-NMR in the comparison of the EAD and non-EAD patients on postoperative day 7 (T6).
Figure 3. Plasma samples analyzed by LC-MS in electrospray positive ion mode, comparing EAD and nonEAD recipients.
The small population number has been one of the limitations of the study as the recruitment of living donor liver transplantation recipients can be limited. Further larger population size is definitely warranted for future studies. We have included the limitation of the study that read “However, the relatively small number of patients can be one limitation. Although lipidomics appears to be a promising tool in the discrimination of patients with EAD and without EAD, validation in a larger data set is warranted.” (Please see page 22)
Also, the discussion has been revised to more focus on the topic as suggested.
